# Prevalence, Molecular Detection, and Antimicrobial Resistance of *Salmonella* Isolates from Poultry Farms across Central Ethiopia: A Cross-Sectional Study in Urban and Peri-Urban Areas

**DOI:** 10.3390/microorganisms12040767

**Published:** 2024-04-10

**Authors:** Hika Waktole, Yonas Ayele, Yamlaksira Ayalkibet, Tsedale Teshome, Tsedal Muluneh, Sisay Ayane, Bizunesh Mideksa Borena, Takele Abayneh, Getaw Deresse, Zerihun Asefa, Tadesse Eguale, Kebede Amenu, Hagos Ashenafi, Gunther Antonissen

**Affiliations:** 1Department of Pathobiology, Pharmacology and Zoological Medicine, Faculty of Veterinary Medicine, Ghent University, 9820 Merelbeke, Belgium; gunther.antonissen@ugent.be; 2Department of Microbiology, Immunology and Veterinary Public Health, College of Veterinary Medicine and Agriculture, Addis Ababa University, Bishoftu P.O. Box 34, Ethiopia; yonasayele30@gmail.com (Y.A.); yamlaksiraayalkibet@gmail.com (Y.A.); kebede.amenu@aau.edu.et (K.A.); 3Department of Clinical Studies, College of Veterinary Medicine and Agriculture, Addis Ababa University, Bishoftu P.O. Box 34, Ethiopia; tsedykeku@gmail.com (T.T.); zeru2008@gmail.com (Z.A.); 4Department of Animal Production Studies, College of Veterinary Medicine and Agriculture, Addis Ababa University, Bishoftu P.O. Box 34, Ethiopia; tsedalmuluneh19@gmail.com; 5Department of Veterinary Science, School of Veterinary Medicine, Ambo University, Ambo P.O. Box 19, Ethiopia; sisayayane12@gmail.com (S.A.); bizunesh.mideksa@ambou.edu.et (B.M.B.); 6National Veterinary Institute (NVI), Bishoftu P.O. Box 19, Ethiopia; takeletefera99@gmail.com (T.A.); getawderesse2007@gmail.com (G.D.); 7Aklilu Lemma Institute of Pathobiology, Addis Ababa University, Addis Ababa P.O. Box 1176, Ethiopia; tadesse.eguale@aau.edu.et (T.E.); hagos.ashenafi1@aau.edu.et (H.A.); 8International Livestock Research Institute (ILRI), Addis Ababa P.O. Box 5689, Ethiopia

**Keywords:** antimicrobials, central Ethiopia, poultry, isolation, multi-drug resistance, PCR, *Salmonella* Enteritidis, *Salmonella* Typhimurium

## Abstract

A cross-sectional study was conducted to assess the prevalence, molecular detection, and antimicrobial resistance of *Salmonella* isolates within 162 poultry farms in selected urban and peri-urban areas of central Ethiopia. A total of 1515 samples, including cloacal swabs (n = 763), fresh fecal droppings (n = 188), litter (n = 188), feed (n = 188), and water (n = 188), were bacteriologically tested. The molecular detection of some culture-positive isolates was performed via polymerase chain reaction (PCR) by targeting spy and sdfl genes for *Salmonella* Typhimurium and *Salmonella* Enteritidis, respectively. Risk factors for the occurrence of the bacterial isolates were assessed. Antimicrobial susceptibility testing of PCR-confirmed *Salmonella* isolates was conducted using 12 antibiotics. In this study, it was observed that 50.6% of the farms were positive for *Salmonella*. The overall sample-level prevalence of *Salmonella* was 14.4%. Among the analyzed risk factors, the type of production, breed, and sample type demonstrated a statistically significant association (*p* < 0.05) with the bacteriological prevalence of *Salmonella*. The PCR test disclosed that 45.5% (15/33) and 23.3% (10/43) of the isolates were positive for genes of *Salmonella* Typhimurium and *Salmonella* Enteritidis, respectively. The antimicrobial susceptibility test disclosed multi-drug resistance to ten of the tested antibiotics that belong to different classes. Substantial isolation of *Salmonella* Typhimurium and *Salmonella* Enteritidis in poultry and on poultry farms, along with the existence of multi-drug resistant isolates, poses an alarming risk of zoonotic and food safety issues. Hence, routine flock testing, farm surveillance, biosecurity intervention, stringent antimicrobial use regulations, and policy support for the sector are highly needed.

## 1. Introduction

The poultry sector in Ethiopia is undergoing significant changes due to the growing human population, particularly in and around major cities and towns [1]. Poultry production plays a crucial role as a livelihood source, ensuring food security, promoting nutrition, and contributing to the economic development of the country. The sector is deeply integrated in Ethiopian society, with poultry farming being practiced by nearly every household in both rural and urban areas [2,3,4].

The considerable productivity potential of the poultry sector in Ethiopia faces several constraints, including reliance on traditional technologies, a high prevalence of diseases, insufficient availability of inputs (such as good quality feed and veterinary drugs), the limited genetic potential of breeds, and suboptimal management practices [5,6]. In Ethiopia, infectious diseases such as Newcastle disease, Salmonellosis, fowl cholera, coccidiosis, and fowl pox were identified as predominant contributors to high morbidity and mortality across various scales of poultry production [7,8,9,10].

Salmonellosis is one of the most important bacterial diseases in the poultry industry and other avian species, causing heavy economic loss due to lowered productivity and also causing a public health hazard by virtue of zoonoses, which is associated with high medication costs [9,11,12]. Avian Salmonellosis occurs in chickens by host-specific *Salmonella* serovars, such as *Salmonella* Pullorum and *Salmonella* Gallinarum, which cause a typhoid-like systemic disease, or a wide range of non-typhoidal *Salmonella,* mainly *Salmonella* Enteritidis and *Salmonella* Typhimurium, together with serovars such as *S.* Newport, *S*. Heidelberg, *S*. Kentucky, *S*. Infantis, *S.* Concord, *S*. Javiana, etc. [13,14,15,16]. Non-typhoidal Salmonellosis is responsible for undetected illness at the farm level, and following the consumption of poultry meat and eggs, humans acquire infection at the plate end. In particular, the non-typhoidal species *Salmonella* Typhimurium and *Salmonella* Enteritidis are responsible for subclinical Salmonellosis in chickens that can induce human infections [17,18,19,20]. Accordingly, it was estimated that non-typhoidal Salmonellosis causes about 93 million enteric infections and 155,000 fatalities worldwide on an annual basis [21,22]. The non-host specificity of the pathogen, its route of transmission, and the existence of multiple antimicrobial resistances are the main reasons contributing to the majority of non-typhoidal Salmonellosis infections [23]. In developing countries of Africa with poor hygiene, weak biosecurity measures, and no or few food safety regulations, prevailing non-typhoidal Salmonellosis remains a serious public health problem [24,25,26], with an occurrence of 10–100/100,000 new cases per year [27]. In Ethiopia, human Salmonellosis is one of the major diseases. For instance, a pooled prevalence of 57.9% was recorded for non-typhoidal *Salmonella* [25]. Previous studies in Ethiopia also indicated that different *Salmonella* serovars have been detected in various regions of the country and have been isolated from humans, animals, food of animal origin, and their environment [10,28,29,30]. In Ethiopia, as well as Sub-Saharan African countries, the problem of Salmonellosis in poultry, as well as humans, is exacerbated by little or no national epidemiological surveillance, a lack of legislation, and an absence of strict enforcement of regulations and intervention measures to address farm biosecurity practices, public hygiene, and regular screening of individuals handling foodstuffs for public consumption [25,31].

In the past, as well as in the present, poultry Salmonellosis has been prevented and controlled by the use of various types of antimicrobials. Unfortunately, there is an increasing trend in the utilization of antimicrobial drugs for animal production to meet the rising demands for animal-derived products by the human population. For instance, the quantity of antimicrobials utilized is anticipated to double in the BRICS countries, encompassing Brazil, Russia, India, China, and South Africa [32]. Quantitatively, the number of antimicrobials used in the livestock sector worldwide was predicted to be 63,151 tons in 2010, and it is estimated to increase by 67% by 2030, attaining nearly 105,500 tons [33].

The emergence of antimicrobial resistance (AMR) can be attributed to an irrational use of first-line drugs and extensive use of antimicrobial drugs coupled with increased consumption of animal products. Previous studies have underlined the potential horizontal dissemination of AMR bacteria and genes between poultry flocks and farms, as well as the extent of zoonotic transmission through the food value chain [34,35,36,37]. Consequently, in the developed world, such as America and Europe, the majority of *Salmonella* isolates from poultry farms and poultry products were found to be resistant to several antimicrobials [38,39,40,41]. However, in Ethiopia, such consolidated findings are lacking, and thus, further studies are needed.

Despite the contribution of the poultry sector to the national economy of Ethiopia, inadequate and fragmented information is available about the true prevalence, distribution, economics, public health significance, and antimicrobial resistance profiles of the zoonotic *Salmonella* serovars *Salmonella* Enteritidis and *Salmonella* Typhimurium in the poultry sector. Having adequate data will contribute to instituting *Salmonella* control programs that ensure poultry health, combatting the risk of foodborne zoonotic infections, and minimizing the escalating risk of antimicrobial-resistant *Salmonella* in poultry farming. Therefore, the present study aimed at monitoring the *Salmonella* Enteritidis and *Salmonella* Typhimurium status in poultry in central Ethiopia, a poultry-dense region characterized by a high variety of production systems. More specifically, the aim of this research was to evaluate the prevalence and AMR profiles of *Salmonella* Enteritidis and *Salmonella* Typhimurium in chickens and environmental samples collected at poultry farms in four selected areas of central Ethiopia.

## 2. Materials and Methods

### 2.1. Study Area

The present study was conducted in four selected areas of central Ethiopia, namely Adama, Addis Ababa, Debre Birhan, and West of Shaggar City, located within a 130 km of the capital city, Addis Ababa (Figure 1). In all the study areas, climate conditions have a bimodal rainfall trend comprising a long rainy season (June to September), a short rainy season (February to May), and a prolonged dry season from late November to February [42].

Livestock production in the study areas was characterized by mixed farming systems, with poultry, dairy, and small ruminant farming being integrated with certain industrial manufacturing activities as the main source of income. These selected areas were representative of commercial poultry production practices in typical highly populated urban areas and surrounding peri-urban communities. The present study areas were considered purposively due to the availability of a high number of commercial poultry farms as well as high demand for poultry and poultry products.

### 2.2. Study Design

A cross-sectional study was conducted from January 2021 to June 2022 in four selected areas of central Ethiopia. Exotic breed chickens reared for the purpose of egg production (Bovans Brown) and meat production (Cobb-500) and, to some extent, mixed types (Saso) maintained under an intensive management system with a deep litter housing system were considered as the study population. With regard to the Saso breed, in some study areas, they are kept as layers while in others as broilers. About ten percent of the poultry farms had a history of *Salmonella* vaccination and were thus purposively excluded from this study. The study targeted only medium-sized farms, with between 1000 and 5000 chickens per farm.

### 2.3. Sample Size Determination and Sampling Strategies

A total of 1515 samples were collected, including cloacal swabs, fresh fecal droppings, litter samples, and feed and drinking water. The repartition of the different samples collected in the respective study areas is illustrated in Table 1.
microorganisms-12-00767-t001_Table 1Table 1Distribution of sample types and number.Study AreasSample Type and Number* *p* (Expected Prevalence)Cloacal SwabsFresh Fecal DroppingsLitterFeedWaterTotalAdama2603030303038028.8% [30]Addis Ababa1675656565639116.5% [28]Debre Birhan1765252525238450% (No previous work)West of Shaggar City1605050505036019% [43]Total 7631881881881881515
* For all the study areas, the sample size was calculated taking into account expected prevalence (as shown in Table 1), 95% confidence interval, and 5% absolute precision as per the formula given by Thrusfield [44]. However, in order to increase precision, additional samples were added.
N = (Zα_/2_)^2^ × P(1 − P)   d^2^
where

N = sample size required;

d = absolute precision (0.05);

P = expected prevalence.

### 2.4. Sampling

Samples were collected according to the recommendations of OIE [45]. A total of 162 poultry farms from the four selected study sites were considered. More specifically, per poultry house, one sample of 25 g pooled fresh fecal droppings (at least from three droppings taken randomly in different locations in the house) was collected with a sterile spatula into the sterile polypropylene tube. Pooled cloacal swabs (of at least three chickens per house) were collected from randomly selected birds using sterile cotton-tipped swabs. Cotton swabs were moistened in buffered peptone water solution before being inserted in the cloaca by gentle rotation in the cloaca of the birds. Immediately following collection, the cloacal swabs were pooled in 10 mL of sterile buffered peptone water, properly plugged and shaken within test tubes. About 25 g of pooled feed and 100 mL of pooled drinking water samples were collected, respectively, from randomly selected chicken feeders and drinkers throughout the poultry house. Furthermore, pooled five-litter samples weighing 5 g each from different sides on the floor of a poultry house were collected using sterile gloves. Collected samples were transported and stored at 4 °C and were either immediately processed upon arrival at the laboratory or the day after.

### 2.5. Isolation and Identification of Salmonella

*Salmonella* was isolated and identified according to standardized protocols described by the International Organization for Standardization for *Salmonella* detection in food and animal feedstuffs ISO 6579 and OIE [45,46]. All sample types were pre-enriched in 225 mL of Buffered Peptone Water (BPW) (1:9) and homogenized by vortexing for two minutes. Then, all pre-enriched samples were incubated at 37 °C for 18–24 h [47,48]. Subsequently, for selective enrichment, 0.1 mL of well-vortexed pre-enrichment sample was inoculated in 10 mL of Rappaport Vassiliadis Soya Peptone (RVS) broth and incubated at 37 °C for 18–24 h [45].

Afterward, incubation of selective/differential culture was performed by streaking a loopful of suspension from the edge of the turbid growth zone onto Xylose Lysine Deoxycholate (XLD) agar and incubating at 37 °C for 24 to 48 h. After incubation, presumptive *Salmonella* colonies were purified on XLD agar. However, agar plates were incubated for a further 24 h and reexamined if typical *Salmonella* colonies were not present. *Salmonella* colonies on XLD were morphologically identified as red colonies with black centers. Typical *Salmonella* colonies were confirmed through six biochemical tests, more specifically Triple Sugar Iron Agar (TSI), Citrate Utilization, Indole, Methyl-Red and Voges–Proskauer (MR-VP) and Lysine Decarboxylation tests [49,50,51,52,53]. These *Salmonella* broth cultures were sub-cultured at two-week intervals adhering to similar procedures until molecular analysis was performed [53].

### 2.6. Molecular Detection of Salmonella Enteritidis and Typhimurium

The biochemically confirmed *Salmonella* isolates were further characterized as *Salmonella* Enteritidis or *Salmonella* Typhimurium by molecular detection of the *sdf*I gene [54] or *spy* gene [55], respectively. Nearly 50% of randomly selected biochemically positive samples from Adama and Debre Birhan areas were subjected to molecular tests. However, molecular confirmation was only performed on *Salmonella* isolates of Adama and Debre Birhan regions.

#### 2.6.1. DNA Extraction

Extraction of DNA was carried out from Tryptic Soy Broth (Becton Dickinson GmbH, Heidelberg, Germany) and TSB sub-cultured *Salmonella* using a DNeasy Blood and Tissue extraction kit (Qiagen, Dusseldorf, Germany) as per the instructions provided by the manufacturer.

#### 2.6.2. Polymerase Chain Reaction (PCR)

The Polymerase Chain Reaction (PCR) was conducted using a thermal cycler (Applied Bio-systems; Genetic Systems Company, Watsonville, CA, USA) for amplification of the *Salmonella* Typhimurium specific gene (*spy*) with an amplicon size of 401 bp and the *Salmonella* Enteritidis specific gene (*sdf*l) with an amplicon size of 304 bp. The forward and reverse primers set for the *spy* gene were 3′- TTA TTC ACT TTT TAC CCC TGA A- 5′ and 5′- CCC TGA CAG CCG TTA GAT ATT- 3′, respectively. Similarly, for the *sdf*l gene, the primer sets were 3′- TGTGTTTTATCTGATGCAAGAGG- 5′forward and 5′- TGAACTACGTTCGTTCTTCTGG- 3′ reverse. The PCR reaction was standardized in a final volume of 20 µL containing nuclease-free water (3 µL), 5 pmol/µL each forward and reverse primers (2 µL for each), I Q ^TM^ supper mix (10 µL) (Bio-Rad Laboratories, Marnes-la-Coquette, France) containing (Taq DNA polymerase, dNTPs, MgCl_2_ and PCR buffer) and DNA template (3 µL). Likewise, positive control (*Salmonella* positive), extraction control (devoid of template DNA), and negative control (nuclease-free water) were also prepared.

DNA amplification was conducted according to the following reaction conditions: an initial denaturation step at 95 °C for 5 min, followed by 35 cycles of denaturation at 95 °C for 30 s, annealing at 52 °C for 40 s, and extension at 72 °C for 30 s with 7 min final extension at 72 °C and holding temperature at 4 °C until gel electrophoresis is performed.

#### 2.6.3. Gel Electrophoresis DNA Band Visualization

After amplification, the PCR fragments were checked using agarose gel electrophoresis and visualized using UV light. Before gel electrophoresis, agarose gel (1.5%), loading dyes, and molecular markers (100-bp) were prepared based on the manufacturer’s recommendation. The total volume of 10 µL mixture of PCR products and loading dye was loaded on 1.5% agarose gel wells, which were prepared from 1% TAE Buffer (Tris-acetate-EDTA) and agarose powder. Similarly, samples, positive and negative control (10 µL each), and molecular ladders (10 µL, 100 bp) were gently loaded on the separate agarose gel wells (lane). Consequently, the amplified DNA product was electrophoresed at 120 volts for one hour. The migration of DNA bands from the agarose gel was visualized using a UV gel documentation apparatus. The amplicons (bands) size of around 401 bp of the target gene (*spy*) for *Salmonella* Typhimurium and 304 bp of the target gene (*sdfI*) for *Salmonella* Enteritidis were visualized and captured on a UV transluminator. The presence of visible bands at or around the expected size was considered positive, whereas the absence of bands at the expected size was considered negative.

### 2.7. Antimicrobial Susceptibility Test

The disc-diffusion method was employed for antimicrobial susceptibility testing of 10 and 15 PCR-confirmed *Salmonella* Typhimurium and *Salmonella* Enteritidis isolates obtained from Adama and Debre Birhan areas, respectively. Based on the recommendations of the International Committee for Clinical Laboratory Standards [56]. This test was performed on Muller Hinton agar medium. Antimicrobial susceptibility was tested for twelve different antimicrobials, namely ampicillin (AMP: 10 µg), azithromycin (AZM: 15 µg), ceftazidime (CAZ: 30 µg), ciprofloxacin (CIP: 5 µg), chloramphenicol (CHL: 30 µg), erythromycin (ERT), gentamycin (GET: 10 µg), kanamycin (KAN: 30 µg), nalidixic acid (NAL), oxytetracycline (OXT: 30 µg), sulfamethoxazole/trimethoprim (SXT: 25 µg) and tetracycline (TET: 30 µg). Antimicrobials were chosen based on their widespread use for the treatment and/or prevention of *Salmonella* infection in livestock production and human health, as well as their accessibility in local markets [28,57,58]. Of each isolate, four to five well-isolated colonies were transferred with the sterile loop into tubes containing 5 mL of Tryptone soya broth (OXOID, CM129, Oxoid Limited, Hampshire, UK). Then, the broth culture was incubated at 37 °C for 6 h and adjusted to attain a turbidity of 0.5 McFarland standards.

Subsequently, a sterile cotton swab was immersed into the suspension, and the bacteria were swabbed evenly over the surface of the Muller Hinton agar plate. Antibiotic disks from each selected antibiotic were placed on the Muller Hinton Agar plate at least 15 mm apart using sterile forceps to avoid overlapping of the inhibition zone. The plates were incubated at 37 °C for 24 h. The diameter of the clear zone of inhibition was measured using a caliper. The results of the antimicrobial sensitivity test were interpreted as sensitive, intermediate, or resistant, according to the interpretation cut-off points for the susceptibility status of bacterial isolates [56].

### 2.8. Data Management and Analysis

All data collected were entered and saved into a Microsoft Excel spreadsheet and then transferred to STATA Version 12 (Stata Corp., College Station, TX, USA) [59] for statistical analysis. The prevalence of Salmonella isolates was calculated using descriptive statistics, in which the number of positives was divided by the total number of samples and multiplied by 100. Pearson’s Chi-square was utilized to assess the statistical significance of various risk factors with the result of the bacteriological and PCR tests. Fisher’s exact test was used for risk factors with few observations. With the ultimate aim of quantifying the crude and adjusted odds ratio (OR), univariate and multivariable logistic regression analyses were conducted, respectively. Statistical significance was declared whenever a p-value of less than 5% (*p* < 0.05) was attained. With regard to determining the effect of various risk factors on the basis of an OR 95% confidence interval, the significance of the statistical test was assumed whenever the confidence interval excluded one of its values.

## 3. Results

### 3.1. Isolation and Identification of Salmonella Species

A cross-sectional study carried out in poultry farms in urban and peri-urban areas of central Ethiopia disclosed an overall farm-level *Salmonella* species prevalence of 50.6% (82/162). However, Adama (70%) and Debre Birhan (73.1%) scored higher prevalences, and there was no statistically significant difference (χ^2^ 6.3 and *p* > 0.05) between the farm level prevalence and studied areas (Table 2).

The present study revealed the farm-level prevalence of *Salmonella* species on the basis of types of production, breed, and age of animals. Accordingly, broiler farms scored 56.5% (26/46)—a relatively higher prevalence as compared to layers, which was 48.3% (56/116). In terms of breed, the higher prevalence was documented in Cobb at 64.7% (22/34) followed by Saso at 50.0% (19/38) and Bovans Brown at 45.6% (41/90). The farm-level *Salmonella* species prevalence indicated that chickens aged 2–5 months had 55.6% (15/27). However, there were no statistically significant differences (*p* > 0.05) between the farm-level prevalence and purpose of production, breed, and age (Table 2).

Among the types of samples examined, the highest prevalence was recorded from fresh fecal droppings (20.2%), followed by litter (19.7%), cloacal swabs (14.5%), and 8.5% for both feed and water. The difference in the overall sample level prevalence across the different samples examined was statistically significant (*p* < 0.05) (Table 3).

In general, the prevalence was higher in fresh fecal droppings and litter samples of Adama accounting for 50% (15/30) and 43% (13/30), respectively. Relatively, the prevalence was lower in water samples examined from Addis Ababa (1.8%), Debre Birhan (5.8%) and West of Shaggar City (10%). There was a statistically significant difference in the prevalence of *Salmonella* species on the basis of type of sample in both Adama (χ^2^ test 12.5 and *p* < 0.05) and Debre Birhan areas (χ^2^ test 10.6 and *p* < 0.05). However, considering the overall prevalence with respect to the type of sample in central Ethiopia, the difference was statistically significant (χ^2^ test 15.1 and *p* < 0.05) (Table 3).

A total of 1515 samples were obtained from cloacal swabs, fresh fecal droppings, litter, feed, and water samples from 162 poultry farms in four selected areas of central Ethiopia. Accordingly, the overall sample level prevalence of *Salmonella* species was 14.4% (218/1515). The sample level bacteriological prevalence in the specific study areas including Adama, Addis Ababa, Debre Birhan, and West of Shaggar City were 88 (23.2%), 23 (5.9%), 55 (14.3%), and 52 (14.4%), respectively. There was a statistically significant difference in the sample level prevalence of *Salmonella* species and study areas (χ^2^ test 46.7 and *p* < 0.001). Types of production and breed were risk factors with a statistically significant difference in the bacteriological prevalence of *Salmonella* species (Table 4).

### 3.2. Molecular Detection of Salmonella Typhimurium and Salmonella Enteritidis

Molecular detection of *Salmonella* Typhimurium and *Salmonella* Enteritidis was conducted on a total of 76 randomly selected cultures, after biochemical confirmation (43 from Adama and 33 from Debre Birhan), which represents a selection of nearly 50% of the bacteriologically confirmed isolates from both study areas. The findings showed the highest molecular detection of *Salmonella* Typhimurium among isolates originated from fresh fecal droppings at 58.3% (7 out of 12), followed by litter at 50% (3 out of 6), and none for water samples. Similarly, the PCR test disclosed that 23.3% (10/43) of the isolates were positive for the *Salmonella* Enteritidis specific gene (*SdfI* gene) (Appendix A) and Table 5). The findings indicated that the highest molecular detection of *Salmonella* Enteritidis was among isolates that originated from fresh fecal droppings at 37.5% (3/8), followed by cloacal swabs at 24% (6/25) and litter at 25% (1/4). However, both water and feed samples were negative for *Salmonella* Enteritidis (Table 5).

In the multivariable logistic regression analysis, age of chickens (2–5 and >6 months) and sample type (fresh fecal droppings and litter) were statistically significantly associated with the bacteriological isolation and molecular detection of *Salmonella* at a *p*-value of <0.05. Accordingly, multivariable logistic regression analysis revealed 4.8 times higher likelihood of bacteriologically isolating *Salmonella* species in chickens with an age of >6 months (*p* < 0.05). Similarly, the odds of isolating *Salmonella* species was 2.3 times (*p* < 0.05) among chickens with ages of 2–5 months (Table 6). Moreover, fresh fecal droppings had higher bacteriological isolation of *Salmonella* species than cloacal swab samples [OR for fresh fecal droppings vs. cloacal swab = 1.87; 95% CI for OR = 1.22–2.88; *p* < 0.05]. Similarly, litter had higher bacteriological isolation of *Salmonella* species than cloacal swab samples [OR for litter vs. cloacal swab = 1.81; 95% CI for OR = 1.17–2.79; *p* < 0.05]. Likewise, molecular detection of *Salmonella* Typhimurium and *Salmonella* Enteritidis was 1.51 and 1.27 times higher (*p* < 0.05) in fresh fecal droppings and litter sample types, respectively (Table 6).

### 3.3. Antimicrobial Susceptibility Profile of Salmonella Typhimurium and Salmonella Enteritidis

Antimicrobial susceptibility testing performed on a total of 15 PCR-positive *Salmonella* Typhimurium isolates obtained from Debre Birhan, and 10 PCR-positive *Salmonella* Enteritidis isolates obtained from Adama, central Ethiopia, indicated that all were resistant or intermediately resistant to two or more of the antimicrobials. The level and extent of resistance of *Salmonella* Typhimurium isolates were the highest for ampicillin (93.3%), followed by oxytetracycline (86.7%) and sulfamethoxazole/trimethoprim (46.7%). Similarly, the *Salmonella* Typhimurium isolates showed 40% resistance each for tetracycline, kanamycin, and erythromycin. On the other hand, *Salmonella* Typhimurium isolates exhibited 100% susceptibility to only two of the twelve antibiotics tested, namely ceftazidime and ciprofloxacin. The majority of *Salmonella* Typhimurium isolates were relatively susceptible to azithromycin (86.7%) and gentamycin (73.3%). The *Salmonella* Enteritidis isolates revealed the highest resistance for ampicillin (90%), followed by 80% resistance against oxytetracycline and tetracycline and nalidixic acid (70%). On the contrary, *Salmonella* Enteritidis isolates showed 100% susceptibility to ceftazidime and ciprofloxacin. The majority of *Salmonella* Enteritidis isolates were relatively susceptible to azithromycin (90%) and chloramphenicol and gentamycin (80%) (Table 7).

In this study, multi-drug resistance patterns were clearly demonstrated in the different *Salmonella* Typhimurium and *Salmonella* Enteritidis isolates. Out of 15 tested *Salmonella* Typhimurium isolates, 26.7%, 20%, and 20.0% exhibited resistance to two, three, and four antibiotics, respectively. However, even two isolates show resistance against eight different antimicrobials (Table 8).

Similarly, the current study showed a multi-drug resistant profile of PCR positive *Salmonella* Enteritidis isolates, exhibiting nine different resistant patterns. Accordingly, out of nine multi-drug resistant isolates, 11.1%, 33.3%, 33.3%, 11.1%, and 11.1% revealed resistance to three, four, five, six and nine antibiotics, respectively (Table 9).

## 4. Discussion

The findings of the present cross-sectional study, for the first time, indicated a higher overall farm-level prevalence of *Salmonella* species, accounting for 50.6% (82/162) in poultry farms situated in urban and peri-urban areas of central Ethiopia. The present finding of 50.6% of farm-level *Salmonella* prevalence was higher than a report from Iran at 36.4% [60], Algeria at 34.37% [48] and Uganda at 20.7% [61]. Low levels of farm prevalence of *Salmonella* were recorded from developed parts of the world, e.g., Denmark at 1.8% [62], Poland at 1.57% [63], and France at 8.6% [64]. On the other hand, the current farm-level prevalence of *Salmonella* was in agreement with Nigeria at 47.9% [65] and Vietnam at 46.3% [66]. The occurrence of low levels of *Salmonella* from European and developed countries can be linked to the application of specific control programs [67], which are deficient and irregular in developing countries like Ethiopia. The high Salmonella prevalence in the present research was attributed to the most likely contributing factor that all the poultry farms were medium-scale carrying thousands of chickens and the husbandry practices associated with intensification permit easy propagation of the bacteria within the farm. In addition, the level of biosecurity implementation in poultry farms in urban and peri-urban parts of central Ethiopia was highly compromised, favoring the occurrence of various diseases, including *Salmonella* [26,68].

The findings of the current study showed no statistically significant differences between the farm-level prevalence of *Salmonella* and the examined risk factors (study area, purpose of production, breed, and age). The widespread occurrence of the bacteria, as well as the relaxation of biosecurity practices in central Ethiopia, contributes to almost equal exposure to the pathogen [26,69,70]. In addition, all the studied farms had uniformity in terms of farm size being medium scale and the production system involving intensive management with a deep litter housing system. This might have resulted in narrow differences in the prevalence of *Salmonella* in the above-mentioned risk factors.

Based on the traditional culture method, *Salmonella* species was identified in 14.4% of the 1515 samples collected from the different selected areas of central Ethiopia. This finding was relatively consistent with earlier reports from Ethiopia at around 15% [68,71,72]. On the contrary, our findings were lower than previous studies carried out in the USA at 38.8% [73], in India at 55% [74], Bangladesh at 31.25% [75], and Uganda at 20.7% [61]. This finding is greater than previous studies [76,77], reporting 2.98% and 9.27% in Jimma and Kefa of southwestern Ethiopia, respectively, and 9.84% in Morocco [78]. However, the lowest level of occurrence of *Salmonella* was observed in European countries, accounting for 2.34% [79,80]. The observed variations in the sample and farm level prevalence of *Salmonella* species could be attributed to factors such as poultry management practices, the housing system of chicken, discrepancies in the biosecurity status, absence of strict disease control programs, scale of farms and hygienic conditions, and intermittent shedding of Salmonellosis [81]. In connection to this, all farms investigated in the present study had deep litter housing systems, and the disregard for sanitary settings might favor widespread *Salmonella* infection [82,83].

The findings of the present study disclosed three risk factors (namely age, breed, and sample type) having a statistically significant association with bacteriological isolation of *Salmonella* species. Consequently, the odds of isolation of *Salmonella* species were 4.98 times and 2.39 times (*p* < 0.05) among chickens with ages of >6 and 2–5 months, respectively. This implies that the chance of acquiring *Salmonella* from the environment increases with age. This is in agreement with earlier studies from Bangladesh [84]. On the contrary, a higher prevalence of *Salmonella* in young chicks was reported from Iran [60]. The observed discrepancies might be attributed to disparities in the production process, isolation technique, and variation in the level of biosecurity practices.

Although differences in the sample prevalence of *Salmonella* species were observed in the different breeds (21.1% for Cobb 500; 12.7% for Saso and 12% for Bovans Brown), it is difficult to establish if susceptibility is linked to genetic variations. More importantly, the farm management system, production types, and level of biosecurity practices could potentially contribute to the differences.

Regarding the isolation and identification of *Salmonella* species on the basis of sample type, the prevalence showed variability being higher in fresh fecal droppings (20.2%) followed by litter (19.7%) and cloacal swabs (14.5%). However, consistent findings were noted in the case of feed (8.5%) and water (8.5%) samples. The results of the present study were higher than earlier reports from Modjo, central Ethiopia, where the isolation from cloacal swabs, feces, litter, and feed accounted for 0.3%, 5.5%, 3.4%, and 0% [30] and 15.2% from feces from six countries of Latin America [85]. On the other hand, this finding is in line with previous studies conducted in southern Ethiopia, which revealed 14.8% in cloacal swabs [71], and Nigeria, which revealed 23% in feces and 20.3% in litter [86]. Isolation of *Salmonella* isolates from fresh fecal droppings (20.2%) and litter (19.7%) in the present study is much lower than the 59.1% reported from poultry litter samples in Nigeria [87] and fresh fecal droppings (92%) from Spain (46.09%) [88]. Similarly, reports from Bangladesh showed higher findings with respect to cloacal swabs (46.09%), feed (18.75%) and water (17.19%) [75]. Likewise, a previous report from Egypt disclosed that 55% of cloacal swabs were positive for *Salmonella* species [89]. The higher isolation of *Salmonella* species in fecal droppings might be due to the fact that the gastrointestinal system is thought to be a potential source of contamination during the intermittent shedding of the pathogen with feces from carrier chickens [90,91]. The detection of *Salmonella* in water and feed, despite lower concentrations, signals a risk of bacterial contamination due to inadequate hygiene management at the farm level, as well as within the feed supply chain. This underscores the importance of implementing robust hygiene practices in poultry operations. Water sources, particularly well water or surface water, can become contaminated right from the supply or through bacterial transmission from the chickens themselves [92].

In alignment with the findings of the current study, globally, *Salmonella* Typhimurium and *Salmonella* Enteritidis are the most predominant isolates responsible for foodborne infections resulting from the consumption of contaminated poultry products in the past couple of decades [17,55,93,94]. *Salmonella* Typhimurium is one of the most threatening serotypes of public health importance and is commonly associated with antibiotic resistance [95]. The 50% detection of *Salmonella* Typhimurium by PCR test in the present study was found to be slightly comparable to 48.9% from Vietnam [96], 46.4% from South Africa [95], 43.35% from Morocco [97], and 40% from Greece [98]. On the contrary, this finding is a lot higher than that of Iran [99], Turkey [100], Singapore [101] and Egypt [102], who reported 1.6%, 9.4%, 18.1%, and 33%, respectively. The results from this study revealed that the existence of *Salmonella* in poultry farms was affected by numerous risk factors and that deep litter systems favor the persistence of *Salmonella* and higher chances of infection [65,103]. The level of application of biosecurity practices could also significantly contribute to the observed variation in different countries [26,104]. *Salmonella Typhimurium,* being the most prevalent isolate in poultry, is well known for its capacity to infect a wide range of animals and for its ability to survive in the environment for long periods, making it one of the most common causes of Salmonellosis [105,106].

The gene *sdf*I was reported to be found only in *Salmonella* Enteritidis and designed as a strong marker for these *Salmonella* serovars [107]. The level of molecular detection of *Salmonella* Enteritidis from the present study was 23.8%. This finding is consistent with the previous reports from Pakistan at 23.3% [10] and Turkey at 21.9% [100], whereas the current finding is much greater than the earlier reports of 7.1% from Ethiopia [108] and 13% from South Africa [109]. The high isolation of *Salmonella* Enteritidis may be because it is more invasive than other serotypes. However, no statistically significant association was observed between the level of molecular detection of *Salmonella* Typhimurium and *Salmonella* Enteritidis and all the risk factors considered in the current study. This might be due to the small number of samples subjected to PCR tests.

The findings of the antimicrobial susceptibility test disclosed that all PCR-confirmed *Salmonella* Typhimurium and *Salmonella* Enteritidis isolates were 100% susceptible to Ceftazidime and Ciprofloxacin. On the contrary, all *Salmonella* Typhimurium and *Salmonella* Enteritidis were noted to express over 80% resistance to ampicillin, oxytetracycline, and tetracycline among the tested antimicrobials. This finding coincides with previous studies carried out in Ethiopia, which reported resistance of *Salmonella* isolates to ampicillin of 97.8% [71] and 100% [110,111]. Interestingly, there has been a practice of extensive antimicrobial usage in poultry farms in central Ethiopia [58]. A recent study carried out in Ethiopia revealed tetracyclines, aminoglycosides, and trimethoprim-sulfonamides were frequently used classes of antibiotics [112].

In contrast to the present study, previous reports from Uganda [61] and Bangladesh [113] revealed that 50.0% and 100% of the *Salmonella* isolates were resistant to ciprofloxacin, respectively. The high level of resistance observed to nine tested antimicrobials is alarming, suggesting that critical antibiotic classes are becoming less effective. This poses challenges in selecting suitable drugs for treating bacterial diseases in poultry. This finding sheds light on the potential consequences of indiscriminate antimicrobial use in poultry farming. The current scenario of antimicrobial resistance is increasingly recognized as a global issue affecting both human and animal health. A key contributing factor to bacterial resistance is the extensive use of antimicrobials in both animal farming and human medicine, coupled with insufficient advocacy and monitoring of antimicrobial utilization [58,112,113,114,115,116]. The use of antimicrobials without prescription and improper dispensing might favor selection pressure that increases the maintenance of resistance genes in bacteria [117]. The present study revealed a multi-drug resistant profile of PCR-confirmed *Salmonella* Typhimurium and *Salmonella* Enteritidis isolates. Accordingly, nine of the eleven tested antimicrobials demonstrated 12 different resistance patterns. These isolates exhibited resistance to two to nine different antibiotics. Such widespread and high degrees of multi-drug resistance have also been demonstrated in other developing countries, such as Egypt [118], Ghana [119], Nigeria [120], Uganda [61], and Senegal [121].

Among other factors, the unregulated access to antimicrobials and the prophylactic use of these drugs starting from day-old chicks can contribute to the selection pressure for resistant isolates. In most instances, broad-spectrum antibiotics are widely used for the treatment of infectious diseases, including Salmonellosis. Such frequent and long-term use of broad-spectrum antibiotics is noted to favor the development of antimicrobial resistance in Ethiopia [30,71], Nigeria [122], and China [123]. In most parts of Africa, including Ethiopia, farmers are absolutely free to use antimicrobials for treatment as well as prophylactic purposes to their perceived benefit [124,125,126]. The development of antimicrobial resistance in Ethiopia is further exacerbated by the practices of accessibility of antibiotics without a valid prescription and without performing the required diagnostic tests. Such indiscriminate use of antimicrobials along with the absence of rational drug use policy and strict regulations greatly contribute to the emergence of resistance [112,127].

## 5. Conclusions

The significant isolation and identification of *Salmonella* Typhimurium and *Salmonella* Enteritidis in poultry and on poultry farms in selected areas of central Ethiopia, coupled with the emergence of multi-drug-resistant profiles as revealed by our study, underscore the urgent need for interventions and public engagement initiatives within the sector. This is particularly critical given the zoonotic potential of these *Salmonella* species. To effectively manage poultry Salmonellosis, it is imperative to implement practical control strategies that enhance biosecurity measures at various production stages. Additionally, ongoing efforts to raise awareness and provide training for farmers and farm workers on the risks of zoonotic diseases and the transmission of antimicrobial-resistant isolates are highly recommended. Establishing stringent and judicious drug use policies, along with interventions to curb the indiscriminate use of antimicrobials, is essential.

## Figures and Tables

**Figure 1 microorganisms-12-00767-f001:**
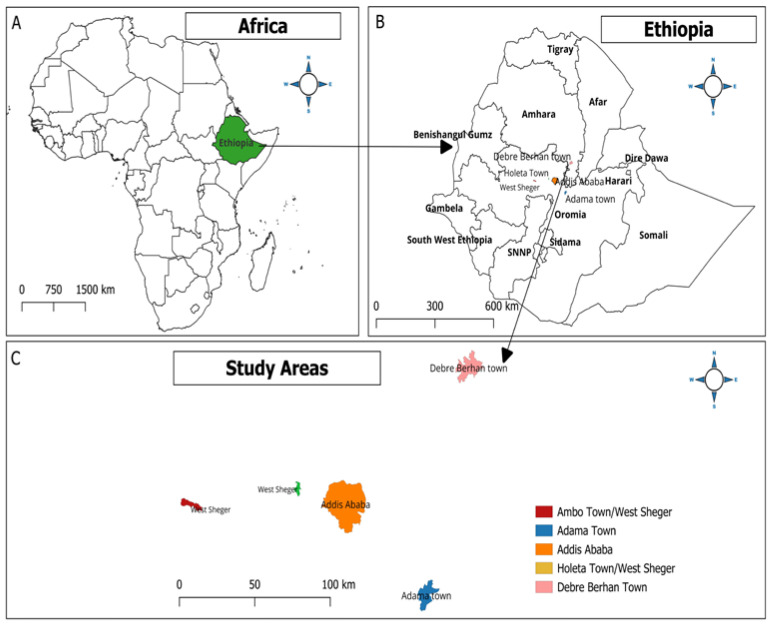
Map showing the study areas using QGIS Ver. 3.14 (QGIS Development Team, 2009. QGIS Geographic Information System. Open-Source Geospatial Foundation. http://qgis.org (accessed on 19 February 2024).

**Table 2 microorganisms-12-00767-t002:** The isolation of *Salmonella* species on the basis of poultry farms examined in selected areas of central Ethiopia.

Risk Factors	Number Farms Tested	Number of Positives	Prevalence (%)	Chi-Square Value (*p*-Value)
Study site				6.3 (0.098)
Adama	30	21	70.0%	
Addis Ababa	56	17	30.4%	
Debre Birhan	26	19	73.1%	
West of Shaggar City	50	25	50.0%	
Overall	162	82	50.6%	
Type of production				0.3 (0.592)
Broiler	46	26	56.5%	
Layer	116	56	48.3%	
Overall	162	82	50.6%	
Breed				1.1 (0.570)
Bovans brown	90	41	45.6%	
Cobb 500	34	22	64.7%	
Saso	38	19	50.0%	
Overall	162	82	50.6%	
Age animals				0.1 (0.950)
<2 months	50	25	50.0%	
2–5 months	27	15	55.6%	
>6 months	85	42	49.4%	
Overall	162	82	50.6%	

**Table 3 microorganisms-12-00767-t003:** Association of different sample types with *Salmonella* prevalence in different regions in central Ethiopia based on routine bacteriological tests.

Study Areas	Cloacal Swab	Fresh Fecal Droppings	Litter	Feed	Water	χ^2^Test	*p* Value
Adama	18.5% (48/260)	50% (15/30)	43% (13/30)	16.7% (5/30)	23.3% (7/30)	12.5	0.014
Addis Ababa	7.7% (14/167)	7.1% (4/56)	5.4% (3/56)	1.8% (1/56)	1.8% (1/56)	4.9	0.334
Debre Birhan	13.6% (24/176)	25% (13/52)	23.1% (12/52)	5.8% (3/52)	5.8% (3/52)	10.6	0.030
West of Shaggar City	15.6% (25/160)	12% (6/50)	18% (9/50)	14% (7/50)	10% (5/50)	1.3	0.859
Overall	14.5% (111/763)	20.2% (38/188)	19.7% (37/188)	8.5% (16/188)	8.5% (16/188)	15.1	0.005

Parentheses indicate the number of positives out of the total examined.

**Table 4 microorganisms-12-00767-t004:** Association of different factors with *Salmonella* prevalence in different regions in central Ethiopia based on routine bacteriological tests.

Risk Factors	Number Samples Tested	Number of Positives	Prevalence (%)	Chi-Square Value (*p*-Value)
Study site				46.7 (*p* < 0.001)
Adama	380	88	23.2	
Addis Ababa	391	23	5.9	
Debre Birhan	384	55	14.3	
West of Shaggar City	360	52	14.4	
Overall	1515	218	14.4	
Type of production				13.2 (*p* < 0.001)
Broiler	492	94	19.1	
Layer	1023	124	12.1	
Overall	1515	218	14.4	
Breed				18.2 (*p* < 0.001)
Bovans brown	793	95	12	
Cobb 500	369	78	21.1	
Saso	353	45	12.7	
Overall	1515	218	14.4	
Age animals				3.5 (*p* = 0.175)
<2 months	296	34	11.5	
2–5 months	517	84	16.2	
>6 months	702	100	14.2	
Overall	1515	218	14.4	

**Table 5 microorganisms-12-00767-t005:** Molecular detection of *Salmonella* Typhimurium and *Salmonella* Enteritidis on the basis of the type of samples examined from Debre Birhan and Adama, central Ethiopia.

Sample Type	No. Samples Tested for		PCR Positive	
	*Salmonella* Typhimurium (Debre Birhan)	*Salmonella* Enteritidis (Adama)	*Salmonella* Typhimurium (Debre Birhan)	*Salmonella* Enteritidis (Adama)
Cloacal swab	9	25	4 (44.5%)	6 (24%)
Fresh fecal droppings	12	8	7 (58.3%)	3 (37.5%)
Litter	6	4	3 (50%)	1 (25%)
Feed	3	3	1 (33.3%)	0 (0%)
Water	3	3	0 (0%)	0 (0%)
Total	33	43	15 (45.5%)	10 (23.3%)
Chi-square value (*p*-value)			4.2 (*p* = 0.041)	

**Table 6 microorganisms-12-00767-t006:** Multivariable logistic regression analysis of *Salmonella* with various risk factors in central Ethiopia.

Risk Factors	Bacteriological Test			PCR Test		
	Number of Positive	Crude Odds Ratio (95% CI)	Adjusted Odd Ratio (95% CI)	Number of Positive	Crude Odds Ratio (95% CI)	Adjusted Odd Ratio (95% CI)
Study site						
Adama	88	Ref	Ref	10	Ref	-
Addis Ababa	23	0.21 (0.13–0.34)	0.18 (0.08–0.43)	-	-	-
Debre Birhan	55	0.55 (0.38–0.80)	0.65 (0.23–1.86)	15	2.75 (1.03–7.36)	-
West of Shaggar City	52	0.56 (0.38–0.82)	0.55 (0.19–1.58)	-	-	-
Type of production						
Broiler	94	Ref	Ref	10	Ref	-
Layer	124	0.58 (0.44–0.78)	0.74 (0.21–2.53)	15	2.75 (1.03–7.36)	-
Breed						
Bovans brown	95	Ref	Ref	15	Ref	Ref
Cobb 500	78	1.97 (1.42–2.74)	3.32 (1.11–9.96)	7	0.24 (0.08–0.72)	0.23 (0.07–0.75)
Saso	45	1.07 (0.73–1.57)	0.91 (0.54–1.52)	3	0.43 (0.09–1.98)	0.33 (0.07–1.64)
Age group						
<2 months	34	Ref	Ref	3	Ref	-
2–5 months	84	0.67 (0.44–1.03)	2.39 (0.94–6.05)	7	1.55 (0.33–7.41)	-
>6 months	100	0.86 (0.62–1.17)	4.98 (2.09–11.87)	15	4.44 (1.48–13.35)	-
Sample type						
Cloacal Swab	111	Ref	Ref	10	Ref	Ref
Fresh fecal droppings	38	1.49 (0.99–2.24)	1.87 (1.22–2.88)	10	2.40 (0.76–7.55)	1.51 (0.43–5.31)
Litter	37	1.44 (0.95–2.17)	1.81 (1.17–2.79)	4	1.60 (0.37–6.92)	1.27 (0.27–6.02)
Feed	16	0.55 (0.32–0.95)	0.65 (0.37–1.15)	1	0.48 (0.05–4.65)	0.31 (0.03–3.39)
Water	16	0.55 (0.32–0.95)	0.65 (0.37–1.15)	0	-	-

Key: ‘Ref’ is the reference category used (odds ratio = 1); ‘-’ analysis not computed.

**Table 7 microorganisms-12-00767-t007:** Antimicrobial susceptibility profile of *Salmonella* Typhimurium and *Salmonella* Enteritidis, central Ethiopia.

Antimicrobial Used	Disc Concentration (µg)	*Salmonella* Typhimurium (15 Selected Samples)			*Salmonella* Enteritidis (10 Selected Samples)		
		No. and % Susceptible	No. and % Intermediate	No. and % Resistant	No. and % Susceptible	No. and % Intermediate	No. and % Resistant
Ampicillin (AMP)	10	0 (0%)	1 (6.7%)	14 (93.3%)	1 (10%)	0 (0.0%)	9 (90%)
Azithromycin (AZM)	15	13 (86.7%)	0 (0.0%)	2 (13.3%)	9 (90%)	0 (0.0%)	1 (10%)
Ceftazidime (CAZ)	30	15 (100%)	0 (0.0%)	0 (0.0%)	10 (100%)	0 (0.0%)	0 (0%)
Ciprofloxacin (CIP)	5	15 (100%)	0 (0%)	0 (0%)	10 (100%)	0 (0%)	0 (0%)
Chloramphenicol (CHL)	30	10 (66.7%)	2 (13.3%)	3 (20%)	8 (80%)	0 (0.0%)	2 (20%)
Erythromycin (ERT)	15	9 (60%)	0 (0%)	6 (40%)	7 (70%)	0 (0%)	3 (30%)
Gentamycin (GNT)	10	11 (73.3%)	4 (20%)	0 (0%)	8 (80%)	2 (20%)	0 (0%)
Kanamycin (KAN)	30	7 (46.7%)	2 (13.3%)	6 (40%)	5 (50%)	2 (20%)	3 (30%)
Nalidixic acid (NAL)	30	8 (53.3%)	4 (26.7%)	3 (20%)	2 (20%)	1 (10%)	7 (70%)
Oxytetracycline (OXT)	30	0 (0%)	2 (13.3%)	13 (86.7%)	0 (0%)	2 (20%)	8 (80%)
Sulfamethoxazole/Trimethoprim (SXT)	25	5 (33.3%)	3 (20%)	7 (46.7%)	6 (60%)	0 (0.0%)	4 (40%)
Tetracycline (TET)	30	6 (40%)	3 (20%)	6 (40%)	2 (20%)	0 (0.0%)	8 (80%)

**Table 8 microorganisms-12-00767-t008:** Multiple antimicrobial resistance patterns of *Salmonella* Typhimurium isolates (n = 15).

Number of Antimicrobials	Resistant Pattern (Number of Isolates)	Proportion of Resistant Isolates
Two	AMP-OXT (3); AMP-KAN (1)	4 (26.7%)
Three	AMP-OXT-SXT (1); AMP-ERT-GNT (1); AMP-OXT-TET (1)	3 (20%)
Four	AMP-OXT-CHL-SXT (1); AMP-CHL-ERT-SXT (1); AMP-OXT-TET-SXT (1)	3 (20%)
Five	AMP-OXT-ERT-TET-SXT (1)	1 (6.7%)
Six	AMP-AZM-TET-KAN-ERT-NA (1); AMP-KAN-GNT-OXT-SXT-NAL (1)	2 (13.3%)
Eight	AMP-AZM-CHL-ERT-KAN-TET-SXT-NAL (2)	2 (13.3%)

AMP = Ampicillin; AZM = Azithromycin; CHL = Chloramphenicol; ERT = Erythromycin; GNT = Gentamycin; KAN = Kanamycin; NAL = Nalidixic Acid; OXT = Oxytetracycline; SXT = Sulfamethoxazole; TET = Tetracycline.

**Table 9 microorganisms-12-00767-t009:** Multiple antimicrobial resistance patterns of *Salmonella* Enteritidis isolates (n = 9).

Number of Antimicrobials	Antimicrobial Resistance Pattern (No.)	Proportion of Resistance Isolates
Three	AMP-NAL-TET (1)	1 (11.1%)
Four	AMP-ERT-SXT-TET (1) AMP-ERT-NAL-TET (1) AZM-ERT-NA-TET (1)	3 (33.3%)
Five	AMP-ERT-NAL-SXT-TET (1) AMP-CHL-AZM-SXT-TET (1) AMP-CHL-KAN-TET-SXT (1)	3 (33.3%)
Six	AMP-CHL-KAN-NAL-SXT-TET (1)	1 (11.1%)
Nine	AMP-AZM-CHL-ERT-KAN-NAL-OXT-SXT-TET (1)	1 (11.11%)

## Data Availability

The data in the present study can be shared upon legitimate request of the corresponding author.

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
