# Peer review of "Prevalence, Molecular Detection, and Antimicrobial Resistance of Salmonella Isolates from Poultry Farms across Central Ethiopia: A Cross-Sectional Study in Urban and Peri-Urban Areas"

_microorganisms, 2024, doi:10.3390/microorganisms12040767_

Round 1

Reviewer 1 Report

Comments and Suggestions for Authors

Comments on the Quality of English Language

Author Response

Dear Reviewer,

Greetings.

We thank you so much for the crucial comments. As per the comments, I hereby kindly attach the response to the comments as one file.

Kind Regards,

Reviewer 2 Report

Comments and Suggestions for Authors

The study by Waktole and colleagues reports the prevalence of Salmonella on poultry farms located in central Ethiopia. Salmonella was detected in around 50% of the farms analyzed. Although this is an alarming result, some issues in the manuscript need to be clarified.

Overall, the manuscript is very long, which makes it confusing to understand at some points. The methodology and presentation of results are the topics that need more attention.

1-The aim of the study should be the same both in the abstract and at the end of the introduction. In the abstract, the authors used "molecular characterization", but only a PCR was performed to identify S. Typhimurium and S. Enteritidis in the study.

2-Molecular detection

Why was this analysis not performed on all isolates? The authors present a table with these data and those obtained by phenotypic (microbiological) identification. But no comparative analysis should be carried out, as in the molecular identification only two regions were included and with samples selected at random, as described by the authors.

Methodology (2.6) can be summarized. There is no need to present Table 2. Authors should only present the final concentrations of each reagent used and the amplification conditions. The electrophoresis conditions and result analysis should also be summarized.

Results: Figures 2 and 3 are not necessary. If the authors wish, they can keep it as supplementary material. Table 6 is confusing. For each region were only the serovar specified in the table analyzed?

3-Antimicrobial susceptibility test

Again, why was the antimicrobial susceptibility profile determined only for PCR-positive isolates?

I know that performing all tests on all isolates may be limited by the availability of financial resources. Therefore, authors should have more rigorous selection criteria to carry out analysis selections.

4-Tables: Please check the need for all tables, or whether it is possible to group information. Also check that all Tables have been identified in the text, with the correct numbering.

5-Discussion

The discussion is limited to comparing the prevalence rates identified in the study with others already described in the literature. Thus, one way to summarize would be to present Results and Discussion together. It is also important to compare data from studies that were carried out in a similar period to the study. References from 2010 and 2013, among others, were used for this topic.

6-Please revise the English language.

7-Minor comments

Please type the scientific names of bacteria and bacterial genes according to the nomenclature standards.

Line 199: please remove “broth” and “kit”

Comments on the Quality of English Language

Moderate editing of English language required.

Author Response

(The authors gave the same response as above.)

Reviewer 3 Report

Comments and Suggestions for Authors

The importance of Salmonella genus is undoubted nowadays, the zoonotic risk overcoming sometimes the economic prejudice of its presence in animals. The risk is further enhanced by the MDR trait of the bacteria. The authors thus tackle an important subject in infectious diseases and preventive medicine - but the use of their results is not the best. There is  consistent duplication of the results in the Results segment (text and tables) and also in the Discussions section, which has to be corrected.

Please see further comments below. 

Introduction: some areas are written in grey colour

Line 25the correct way of writing is Salmonella (italics) followed by Typhimurium (T in upper case), please correct in all instances

Line 33: maybe it would be interesting to mention in the sentence the classes (multi-drug resistance to ten of the tested antibiotics), otherwise vague

Line 48: is “areas” missing? (in both rural and urban [2-4].)

Line 57: species,  Italic again

Lines 59-62: please correct the size of the font in Salmonella

Lines 85-86: “In the past as well as in the present poultry salmonellosis have been prevented and 85 controlled by the use of various types of antimicrobials” (has been prevented?)

Lines 99-100: do not start with capital W

Line 107: Salmonella different font size

Fig 1: West Sheger is in two locations on the map?

Line 125: http://qgis.org - different font size

Line 149: it is Thrusfield (44) not Thursfield

Lines 186-187: Methyl-Red and 186 Voges-Proskauer (MR-VP) and Lysine Decarboxylation tests [49-53 – unequal size and type of font

Lines 194-195: “However, molecular confirmation was only performed on Salmonella isolates of Adama and Debre  Birhan regions.” – reason?

Page 6: empty

Line 239: correct Adam to Adama

Lines 242-246: classes of antibiotics should be identified for a more efficient discussion of MDR

Page 8: half empty

Line 267: was calculated

Move table 3 to page 9

Lines 323-314 repeat Table 5 data

Line 326: same for 3.2. Molecular Detection of Salmonella Typhimurium and Salmonella Enteritidis and Table 6

Lines 350-353: Legend under Fig.3 title

Line 359: Salmonella species – Salmonella different font size and bold, species Italic – please correct

Table 7. Multivariable logistic regression analysis of Salmonella with various risk factors in central 367 Ethiopia – the information is repeated in the text

Line 377: it would be advisable to mention the antibiotic classes in connection with MDR

Lines 377-389: repetition of table 8; the authors should consider to rephrase and explain the results. In connection with MDR, MAR index could be calculated

Line 403: resistance patterns

Table 11: repetition in the text

Page 16 empty space

Discussions: major flaws- repetition of the results, in some places just a review of findings in other countries without critical discussion of the differences, etc.

Line 559Salmonella species; please correct species is not Italic

Comments on the Quality of English Language

none

Author Response

(The authors gave the same response as above.)

Round 2

Reviewer 2 Report

Comments and Suggestions for Authors

Dear authors,

Thank you for all replies.

There are minor corrections that can be made during the proofing stage.

Please, check the spelling of gene names. They must be written in lowercase letters and in italics: spy and sdf1.

Page 6 lines 212-213: please remove capital letter from “amplicon”

Page 6, line 227: please correct “seconds” (40 seconds... 30 seconds)

Page 6, line 228: please remove capital letter from “gel”

Page 7, line 226: please change “hrs” to “hours”

Page 9, Table title: please add “of” – “on the basis of poultry farms”

Page 10, line 323: please remove “respective”

Page 16, line 430: please add “in” – “was in agreement with”

Page 17, line 468: please remove “(Mahmud et al., 2011)”

Page 17, line 472: please remove italic from “species”

Page 17, line 484: please add “that” – “Ethiopia that revealed”

Page 18, lines 518 and 522: please remove “(Agron et al., 2001)”, “(Adamu, 2017)”, “(Ramatla, et al., 2020)”

Author Response

Dear Reviewer, 

Thank you so much for the valuable and constructive comments that improved the quality of the manuscript. We have seen that all the comments were quite appropriate and accommodated accordingly.

Best Regards,
